# Beyond Invariance: Test-Time Label-Shift Adaptation for Addressing "Spurious" Correlations

**Qingyao Sun**[*]
Cornell University
qs234@cornell.edu

**Kevin Murphy**
Google DeepMind
kpmurphy@google.com

**Sayna Ebrahimi**
Google Cloud AI Research
saynae@google.com

**Alexander D'Amour**
Google DeepMind
alexdamour@google.com

## Abstract

Changes in the data distribution at test time can have deleterious effects on the performance of predictive models $p(y|x)$. We consider situations where there are additional meta-data labels (such as group labels), denoted by $z$, that can account for such changes in the distribution. In particular, we assume that the prior distribution $p(y, z)$, which models the dependence between the class label $y$ and the "nuisance" factors $z$, may change across domains, either due to a change in the correlation between these terms, or a change in one of their marginals. However, we assume that the generative model for features $p(x|y, z)$ is invariant across domains. We note that this corresponds to an expanded version of the widely used "label shift" assumption, where the labels now also include the nuisance factors $z$. Based on this observation, we propose a test-time label shift correction that adapts to changes in the joint distribution $p(y, z)$ using EM applied to unlabeled samples from the target domain distribution, $p_t(x)$. Importantly, we are able to avoid fitting a generative model $p(x|y, z)$, and merely need to reweight the outputs of a discriminative model $p_s(y, z|x)$ trained on the source distribution. We evaluate our method, which we call "Test-Time Label-Shift Adaptation" (TTLSA), on several standard image and text datasets, as well as the CheXpert chest X-ray dataset, and show that it improves performance over methods that target invariance to changes in the distribution, as well as baseline empirical risk minimization methods. Code for reproducing experiments is available at `https://github.com/nalzok/test-time-label-shift`.

## 1 Introduction

Machine learning systems are known to be sensitive to so-called "spurious correlations" [Geirhos et al., 2020, Wiles et al., 2022, Arjovsky et al., 2019] between irrelevant features of the inputs and the predicted output label. These features and their associated correlations are called "spurious" because they are expected to change across real-world distribution shifts. As a result, models that rely on such spurious correlations often have worse performance when they are deployed on a *target domain* that is distinct from the *source domain* on which they were trained [Geirhos et al., 2020, Izmailov et al., 2022]. For example, Jabbour et al. [2020] show that neural networks trained to recognize conditions like pneumonia from chest X-rays can learn to rely on features that are predictive of patient demographics rather than the medical condition itself. When the correlation between demographic

---

[*]Work done as a master's student at the University of Chicago.

37th Conference on Neural Information Processing Systems (NeurIPS 2023).

factors and the condition change (e.g., when the model is deployed on a different patient population), the performance of such models suffers.

To address this issue, recent work has focused on learning predictors that are invariant to changes in spurious correlations across source and target domains, either using data from multiple environments [e.g., Arjovsky et al., 2019], or data where the labels for the nuisance factors are available at training time [Veitch et al., 2021, Makar et al., 2022, Puli et al., 2022, Makino et al., 2022].

However, "spurious" correlations can sometimes provide valuable prior information for examples where the input is ambiguous. Consider the example of calculating the probability that a patient has a particular disease given a positive test. It is well known that the underlying prevalence of the disease (i.e., prior probability it is present) in the patient population is highly informative for making this diagnosis; even if the test is highly sensitive and specific, the patient's probability of having the disease given a positive test may be low if the disease is rare given that patient's background [see, e.g., Bours, 2021]. A similar logic applies in many prediction problems: if we can predict, say, patient demographics from the input, and the prevalence of the target label differs between demographic groups in the current patient population, then the demographic-relevant spurious features provide prior information that supplements the information in the target-relevant features of the input. Thus, if a model could be adjusted to use spurious features optimally in downstream target domains, it could substantially out-perform invariant predictors, as we show in this paper.[2]

Motivated by the above, in this paper we propose a test-time approach for optimally adapting to distribution shifts which arise due to changes in the underlying joint prior between the class labels $y$ and the nuisance labels $z$. We can view these changes as due to a hidden common cause $u$, such as the location of a specific hospital. Thus we assume $p_s(u) \neq p_t(u)$, where $p_s$ is the source distribution, and $p_t$ is the target distribution. Consequently, $p_i(y, z) = \sum_u p(y, z|u)p_i(u)$ will change across domains $i$. However, we assume that the generative model of the features is invariant across domains, so $p_i(\boldsymbol{x} \mid y, z) = p(\boldsymbol{x} \mid y, z)$. See Figure 1 for an illustration of our modeling assumptions.

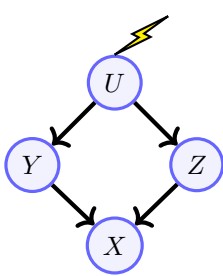

Figure 1: Modeling assumptions. $U$ is a hidden confounder that induces a spurious correlation between the label $Y$ and other causal factors $Z$, which together generate the features $X$. The distribution of $U$ can change between target and source domains, but the distribution of $X$ given $Y, Z$ is fixed.

The key observation behind our method is that our assumptions are equivalent to the standard "label shift assumption", except it is defined with respect to an expanded label $m = (y, z)$, which we call the meta-label. We call this the "expanded label shift assumption". This lets use existing label shift techniques, such as Alexandari et al. [2020], Lipton et al. [2018], Garg et al. [2020], to adapt our model using a small sample of *unlabeled* data $\{\boldsymbol{x}_n \sim p_t(\boldsymbol{x})\}$ from the target domain to adjust for the shift in the prior over meta-labels, as we discuss in Section 3.2. Importantly, although our approach relies on the assumption that $p(\boldsymbol{x} \mid y, z)$ is preserved across distribution shifts, it is based on learning a *discriminative* base model $p_s(y, z, \mid \boldsymbol{x})$, which we adjust to the target distribution $p_t(y \mid \boldsymbol{x})$, as we explain in Section 3.1. Thus we do not need to fit a generative model to the data. We do need access to labeled examples of the confounding factor $z$ at training time, but such data is often collected anyway (albeit in limited quantities) especially for protected attributes. Additionally, because it operates at test-time, our method does not require retraining to adapt the base model to multiple target domains. We therefore call our approach Test-Time Label Shift Adaptation (TTLSA).

We evaluate TTLSA on various standard image and text classification benchmarks, as well as the CheXpert chest X-ray dataset. We show that in shifted target domains TTLSA outperforms ERM (empirical risk minimization, which often uses spurious features inappropriately in target domains), as well as methods that train invariant classifiers (which ignore spurious features). Note, however,

---

[2]We acknowledge that practitioners may have concerns beyond target domain performance, such as certain notions of equal treatment, in which the reliance on such spurious features is problematic or even forbidden. However, equal treatment criteria are often motivated by a desire to ensure that models perform as well as possible when they are applied in domains with very different population compositions, and our method can achieve this goal [see Makar and D'Amour, 2022, for related discussion].

that it has been shown that no single adaptation method can work under all forms of distribution shift [Veitch et al., 2021, Kaur et al., 2022]. Our assumptions capture certain kinds of shift, but certainly not all. In particular, our method is unlikely to help with the kinds of problems studied in the domain adaptation literature, where there is covariate shift (i.e., a change from $p_s(\boldsymbol{x})$ to $p_t(\boldsymbol{x})$) which is not captured by a change in the distribution over the causal factors $(y, z)$.

## 2 Related work

**Spurious correlations, invariant learning, and worst-group performance**    Spurious correlations have mostly been studied in the domain generalization literature (see e.g., [Koh et al., 2020, Gulrajani and Lopez-Paz, 2021]), in which a model is expected to generalize (i.e., achieve acceptable performance) in a target domain without access to any data from that target domain. In this problem setup, building predictors based on the principle of invariance or minimax optimality with respect to spurious correlation shifts is a natural approach.

Many of these methods make the same modeling assumptions as we do (shown in Figure 1); this is often referred to as an anti-causal prediction setting, since the labels cause (generate) the features rather than vice versa [Schoelkopf et al., 2012, Veitch et al., 2021, Makar et al., 2022, Puli et al., 2022, Zheng and Makar, 2022]. These methods use the fact that an invariant predictor will satisfy certain conditional independencies, and employ regularizers that encourage the desired conditional independence. A related line of work tries to minimize the worst-case loss across groups (values of $m = (y, z)$) using distributionally robust optimization [see, e.g., Sagawa et al., 2020, Liu et al., 2021, Nam et al., 2022, Lokhande et al., 2022]). Recently, Idrissi et al. [2022] showed empirically that the "SUBG" method, which uses ERM on a group-balanced version of the data achieved by subsampling, can be an effective approach to learning worst-group-robust predictors in this setting.

**Unsupervised domain adaptation and label shift**    In the Unsupervised Domain Adaptation (UDA) literature, we assume that unlabeled data from the target domain is available, either at training or test time. To make optimality guarantees, UDA methods require assumptions about the structure of the distribution shift. Methods are often categorized into whether they require a covariate shift assumption (i.e., that the discriminative distribution $p(y \mid \boldsymbol{x})$ is preserved [see, e.g., Shimodaira, 2000]), or a label shift assumption (i.e., that generative distribution $p(\boldsymbol{x} \mid y)$ is preserved [Lipton et al., 2018, Garg et al., 2020]).

In the setting of Figure 1, neither the standard covariate shift nor label shift assumptions hold; however, the label shift assumption does hold with respect to the meta-label $m = (y, z)$. Our primary contribution is to show that label shift adaptation techniques, such as Alexandari et al. [2020], when applied with respect to the meta-label, can also be used to adapt to changes due to spurious correlations.

Interestingly, in the "purely spurious" setting described in Makar et al. [2022], which is a special case of Figure 1, we can obtain a risk-invariant model as a special case, by adapting to a target distribution for which $p_t(y, z) = p_s(y)p_s(z)$ [Veitch et al., 2021, Makar and D'Amour, 2022]. [3] Based on this observation, we show that logit adjustment [Menon et al., 2021], a set of test- and training-time methods developed for long-tail learning, can be repurposed to do invariant learning when applied to the meta-label $m = (y, z)$.

**Methods using the expanded label shift assumption**    The "generative multi-task learning" or GMTL method of Makino et al. [2022] makes the same expanded label shift assumption that we do. However, instead of estimating $p_t(y, z)$ from unsupervised target data, they instead assume that there is some value $\alpha$ such that $p_t(y, z) = p_s(y, z)^{1-\alpha}$. They state that choosing $\alpha$ is an open problem, and therefore they report results for a range of possible $\alpha$'s. By contrast, we provide a way to estimate $p_t(y, z)$, and we do not restrict it to have the above functional form.

---

[3]The "purely spurious" term was coined in Veitch et al. [2021], but for brevity we describe using formalism from Makar et al. [2022]. Within the model of Figure 1, an association is "purely spurious" if there is a sufficient statistic of $X$, $e(X)$, such that $Y \perp X \mid e(X), Z$ and $e(X) \perp Z \mid Y$. This occurs when the influence of $Y$ on $X$ can be localized in some features $e(X)$ that are not affected by $Z$, or intuitively, when the influences of $Y$ and $Z$ on $X$ are disentangleable. The results of Idrissi et al. [2022], where spurious correlations are neutralized by data balancing, suggests that pure spuriousness is a common case, at least among worst-group benchmarks. We discuss pure spuriousness, worst-group performance, and invariance in more detail in the supplement.

In Jiang and Veitch [2022], they propose a method called Anti-Causal Transportable and Invariant Representation or "ACTIR" which also makes the same expanded label shift assumption that we do, and further relaxes the assumption that $Z$ is observed during source training. However, they require access to examples from multiple source distributions, which they use to learn a domain invariant classifier. Furthermore, then require labeled $(x, y)$ examples to adapt their classifier at test time.

**Test time adaptation** There is a growing Test Time Adaptation (TTA) literature that explores strategies for adapting a trained model at test time using unlabeled data from the target domain. These methods are based on various heuristics to adapt discriminative models without specifying strong assumptions on the distribution shift structure. For example, TENT [Wang et al., 2021] uses entropy minimization to update the batch normalization layers of a CNN, and MEMO [Zhang et al., 2021] uses ensembles of predictions for different augmentations of a test sample. Most of the TTA literature has focused on models that work on images (e.g., by leveraging data augmentation), so these techniques cannot be applied to embeddings or other forms of input. By contrast, our method can be applied to any classifier, even non-neural ones, such as random forests. For a more comprehensive review of TTA methods, see [Liang et al., 2023].

## 3 Test-Time Label Shift Adaptation

Our aim is to construct a Bayes optimal predictor for the target distribution, $f_t(\boldsymbol{x})$, using a large labeled dataset from the source distribution, $\mathcal{D}_s^{xyz} = \{(\boldsymbol{x}_n, y_n, z_n) \sim p_s(\boldsymbol{x}, y, z)\}$, and a small unlabeled dataset from the target distribution, $\mathcal{D}_t^{\boldsymbol{x}} = \{\boldsymbol{x}_n \sim p_t(\boldsymbol{x})\}$. The optimal prediction for the class label in the target domain is given by

$$f_t(\boldsymbol{x}) = \arg\max_y p_t(y|\boldsymbol{x}) = \arg\max_y \sum_z p_t(y, z|\boldsymbol{x}). \tag{1}$$

where the joint posterior over the class label $y$ and the nuisance factor $z$ is given by

$$p_t(y, z|\boldsymbol{x}) \propto p_t(\boldsymbol{x}|y, z)p_t(y, z) = p_s(\boldsymbol{x}|y, z)p_t(y, z) \tag{2}$$

where the first step follows from Bayes' rule, and the second step follows from our causal stability assumption that $p_t(\boldsymbol{x}|y, z) = p_s(\boldsymbol{x}|y, z) = p(\boldsymbol{x}|y, z)$.

The first key insight of our method is that we can use the EM algorithm (or other optimization methods) to estimate $p_t(y, z)$ by maximizing the likelihood of the unlabeled target dataset, as explained in Section 3.2. The second key insight is that we can estimate the likelihood $p(\boldsymbol{x}|y, z)$ up to a constant of proportionality by fitting a discriminative model on the source dataset, and then dividing out by the source prior:

$$p_s(\boldsymbol{x}|y, z) \propto \frac{p_s(y, z|\boldsymbol{x})}{p_s(y, z)} \tag{3}$$

This is known as the "scaled likelihood trick" [Renals et al., 1994], and avoids us having to fit a generative model. We can estimate $p_s(y, z|\boldsymbol{x})$ using any supervised learning method. However, to get good performance in practice, we need to ensure the probabilities are calibrated, as we explain in Section 3.1. Finally, we can estimate the source prior $p_s(y, z)$ just by counting how often each $(y, z)$ combination occurs in $\mathcal{D}_s^{xyz}$ and then normalizing. Combining Equation (2) with Equation (3) we can compute the posterior distribution over augmented labels from the unlabeled target data as follows:

$$p_t(y, z|\boldsymbol{x}) \propto \frac{p_s(y, z|\boldsymbol{x})}{p_s(y, z)} p_t(y, z). \tag{4}$$

We can then compute $f_t(\boldsymbol{x})$ using (1). In summary, our TTLSA method consists of the following two steps:

1. **Train on source.** Train a model $p_s(y, z|\boldsymbol{x})$ using supervised learning (and calibration) applied to $\mathcal{D}_s^{xyz}$, as explained in Section 3.1. Also estimate $p_s(y, z)$ from $\mathcal{D}_s^{xyz}$ by counting.
2. **Adapt to target.** Estimate $p_t(y, z)$ using EM applied to $\mathcal{D}_t^x$, as explained in Section 3.2. Then compute $p_t(y, z|\boldsymbol{x})$ using (2) and $f_t(\boldsymbol{x})$ using (1).

We discuss these steps in more detail below.

## 3.1 Fit model to source distribution

In the first step, we fit a discriminative classifier $p_s(y, z|\boldsymbol{x})$ on the source dataset. This is just like a standard classification problem, except we use an augmented output space, consisting of the class label $y$ and group label $z$; we denote this joint label by $m := (y, z)$.

**Calibration** Our adaptation procedure crucially relies on the fact that the output of the classifier, $p_s(m|\boldsymbol{x})$, can be interpreted as calibrated probabilities. Since modern neural networks are often uncalibrated [see, e.g., Guo et al., 2017], we have found it important to perform an explicit calibration step. (See the supplementary for an ablation study where we omit the calibration step.) Specifically, we follow Alexandari et al. [2020] and use their "bias corrected temperature scaling" (BCTS) method, which is a generalization of Platt scaling to the multi-class case. In particular, let $l(\boldsymbol{x})$ be the vector of $M$ logits. We then modify $p_s(m|\boldsymbol{x})$ as follows:

$$p_s(m|\boldsymbol{x}) \propto \exp\left(\frac{l(\boldsymbol{x})_m}{T} + b_m\right) \tag{5}$$

where $T \geq 0$ is a learned temperature parameter, and $b_m$ is a learned bias. This calibration is done after the source classifier is trained using a labeled validation set from the source distribution.

**Logit adjustment** The calibration method is a post-hoc method. However, if the source distribution has a "shortcut", based on spurious correlations between $z$ and $y$, the discriminative model may exploit this by overfitting to it. The resulting model will not learn robust features, and will therefore be hard to adapt, even if we use calibration. To tackle this, we take inspiration from Proposition 1 of Makar et al. [2022], which showed that a classifier that is trained on the unconfounded or balanced distribution $p_b$, such that $p_b(y, z) = p_s(y)p_s(z)$, will not learn any "shortcut" between the confounding factor $z$ and the target label $y$. Such a model will therefore have a risk which is invariant to changes in the $p(z|y)$ distribution, and, in "purely spurious" anti-causal settings (a special case of Figure 1, see footnote 3), they showed that this is indeed the optimal one among all risk-invariant predictors (see discussion in Section 2). If both marginals are uniform, then $p_b(y, z) \propto 1$, so the corresponding balanced version of the source distribution becomes $p_b(m|\boldsymbol{x}) \propto \frac{p_s(m|\boldsymbol{x})}{p_s(m)}$, which in log space becomes $\log(p_b(m|\boldsymbol{x})) \equiv \log(p_s(m|\boldsymbol{x})) - \log(p_s(m))$. We train our classifier to maximize this balanced objective; this is known as "logit adjustment" [Menon et al., 2021]. After fitting, we can recover the classifier for the original source distribution using $p_s(y, z|\boldsymbol{x}) \propto p_b(y, z|\boldsymbol{x})p_s(y, z)$. Finally we apply calibration to $p_s(y, z|\boldsymbol{x})$, as explained above. (Note that when the marginals are not uniform, one can apply a similar adjustment, subtracting off $\log(p_s(y, z)/(p_s(y)p_s(z)))$.)

## 3.2 Adapt model to target distribution

Given some unlabeled samples from the target distribution, $\mathcal{D}_t^x = \{\boldsymbol{x}_n \sim p_t(\boldsymbol{x}) : n = 1 : N\}$, our goal is to estimate the new prior, $p_t(y, z) = p_t(m) = \pi_m$. This is a standard subroutine in label shift adaptation methods [Lipton et al., 2018, Garg et al., 2020]; our innovation is to do this with respect to the meta-label $m$. Here, we use an EM approach [Alexandari et al., 2020], and maximize the log likelihood of the unlabeled data, $\mathcal{L}(\boldsymbol{x}; \boldsymbol{\pi}, \boldsymbol{\theta}) = \sum_{\boldsymbol{x}_n} \log p_t(\boldsymbol{x}_n; \boldsymbol{\pi}, \boldsymbol{\theta})$, wrt $\boldsymbol{\pi}$, where

$$p_t(\boldsymbol{x}; \boldsymbol{\pi}, \boldsymbol{\theta}) = \log\left(\sum_m \pi_m p_t(\boldsymbol{x}|m; \boldsymbol{\theta})\right) = \log\left(\sum_m \pi_m \frac{p_s(m|\boldsymbol{x}; \boldsymbol{\theta})}{p_s(m; \boldsymbol{\theta})}\right) + \text{const} \tag{6}$$

where $\boldsymbol{\theta}$ are the parameters estimated from the source distribution. This objective is a sum of logs of a linear function of $\boldsymbol{\pi}$, and is maximized subject to the affine constraints $\pi_m \geq 0$ and $\sum_{m=1}^M \pi_m = 1$. Thus, under weak conditions on the ratios $p_s(m|\boldsymbol{x}; \boldsymbol{\theta})/p_s(m; \boldsymbol{\theta})$ (these implied when $\boldsymbol{x}$ contains some information that can discriminate between levels of $m$) the problem is concave, with a unique global optimum, implying the parameters are identifiable [Alexandari et al., 2020].

A simple way to compute this optimum is to use EM, which automatically satisfies the constraints. Let $\pi_m^j$ be the estimate of $\pi_m$ at iteration $j$. We initialize with $\pi_m^0 = p_s(m)$ and run the following

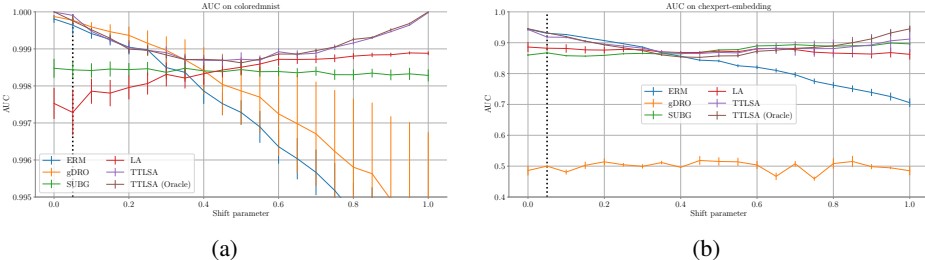

(a)                                                           (b)

Figure 2: Test set AUC performance on (a) Colored MNIST and (b) CheXpert as we vary the correlation between $y$ and $z$ in the target distribution. The vertical dotted line marks the source distribution. There are three performance regimes: (1) the unadapted ERM model (blue line) degrades smoothly under shift; (2) the invariant models (yellow, green, and red lines) have flat performance; and (3) the adapted models (purple and brown lines) out-perform invariant models, yielding a U-shape.

iterative procedure below:

$$p_t(m|\boldsymbol{x}_n; \boldsymbol{\pi}^j, \boldsymbol{\theta}) \propto p_t(\boldsymbol{x}_n|m; \boldsymbol{\theta})\pi_m^j \propto \frac{p_s(m|\boldsymbol{x}; \boldsymbol{\theta})}{p_s(m; \boldsymbol{\theta})}\pi_m^j \text{ // E step} \tag{7}$$

$$\pi_m^{j+1} = \frac{1}{N}\sum_{n=1}^{N} p_t(m|\boldsymbol{x}_n; \boldsymbol{\pi}^j, \boldsymbol{\theta}) \text{ // M step} \tag{8}$$

We then set $p_t(m) = \boldsymbol{\pi}_m^J$. We can also modify this to compute a MAP estimate, instead of the MLE, by using a Dirichlet prior. See the supplementary for a detailed derivation.

After estimating $p_t(y, z)$ on $\mathcal{D}_t^x$, we can compute $p_t(y, z|\boldsymbol{x})$ for the examples in $\mathcal{D}_t^x$ using Equation (4).

## 4   Experiments

In this section, we provide an experimental comparison of our method with various baseline methods on a variety of datasets. In particular, we compare the following methods:

**ERM** This corresponds to training a model on the source distribution, and then applying the same model to the target distribution without any adaptation, i.e., we assume $p_t(y|\boldsymbol{x}) = p_s(y|\boldsymbol{x})$.

**gDRO** The group DRO method of [Sagawa et al., 2020] is designed to optimize the performance of the worst performing group. (We only use this method for the worst-group benchmarks in Section 4.3.)

**SUBG** The "SUBG" method of [Idrissi et al., 2022] subsamples the data so there is an equal number of examples in each group $m = (y, z)$, then trains a classifier by standard ERM. This is a simpler alternative to gDRO yet often achieves comparable performance.

**LA** This is our logit adjustment method of Section 3.1, which approximates a domain invariant classifier by targeting a uniform prior on the source domain, thus avoid overfitting to spurious correlations.

**TTLSA** This is our EM method of Section 3.2 that adapts the LA-based classifier using unlabeled data.

**Oracle** This is similar to TTLSA, but we replace the EM procedure with the ground truth target meta-label prior $p_t(y, z)$. This gives an upper bound on performance. (We only use this for the CMNIST experiments in Section 4.1 and the CheXpert experiments in Section 4.2, where we artificially control the degree of distribution shift.)

### 4.1   Colored MNIST

In this section, we apply our method to the Colored MNIST dataset [Arjovsky et al., 2019, Gulrajani and Lopez-Paz, 2021].

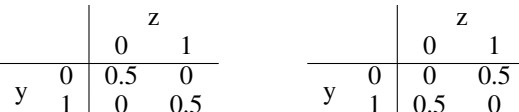

Table 1: The two "anchor" distributions, reflecting total positive and negative correlation between the class label $y$ and the confounding factor $z$. Left: $p_0$. Right: $p_1$. From these distributions, we can create a family of target distributions $p_\lambda = \lambda p_0 + (1 - \lambda)p_1$.

**Dataset**  We construct the dataset in a manner similar to [Arjovsky et al., 2019, Gulrajani and Lopez-Paz, 2021]. Specifically, we create a binary classification problem, where label $y = 0$ corresponds to digits 0-4, and $y = 1$ corresponds to 5-9. We then sample a random color $z \in \{0, 1\}$, corresponding to red or green, for each image, with a probability that depends on the target distribution. See the supplementary material for a visualizaiton of this dataset. Since the color is easier to recognize than the shape, color can act as a "shortcut" to predicting the class label, even though this is not a robust (domain invariant) feature.

We create a set of 21 target distributions $p_\lambda = (1 - \lambda)p_0 + \lambda p_1$, where $\lambda \in \{0, 0.05, \ldots, 0.95, 1.0\}$, and $p_0(y, z)$ and $p_1(y, z)$ are two anchor distributions in which $y$ and $z$ are correlated and anti-correlated, respectively (see Table 1). By changing $\lambda$, we can control the dependence between $y$ and $z$. We train the classifier on a source domain exhibiting a strong spurious correlation (we choose $\lambda = 0.05$), and then apply the model to each target domain, $p_\lambda$, for $\lambda \in \{0, 0.05, \ldots, 0.95, 1.0\}$. We measure performance using Area Under the ROC curve (AUC).

**Training procedure**  During training, we fit a LeNet CNN on the training set by using $m = (y, z)$ as the label. We use AdamW with a batch size of 64 and a learning rate of $10^{-3}$ for a maximum of 5000 epochs, and run calibration for a maximum of 1000 epochs with the same learning rate and batch size on a 10% holdout set. However, training typically terminates much earlier since we also calculate the exponential moving average of the validation loss with a decay rate of 0.1, and stop early when the smoothed validation loss does not decrease for 5 consecutive epochs. During evaluation, we infer the target label prior for each target distribution using EM applied to an unlabeled test set of size 64 or 512, and we then predict the class labels on this test set. Finally we repeat this across all the unlabeled minibatches in the target distribution, and report the overall AUC.

**Results**  Figure 2a shows our results, with error bars showing standard error of the mean across 4 trials. There are three main groups of curves. (1) The ERM model (black line) shows performance that gets steadily worse as the target distribution shifts away from the source. (2) The invariant models (SUBG in dashed orange and logit adjustment in solid red) are approximately constant across domains, as expected. (In this experiment, we see that logit adjustment outperforms SUBG.) 3) Our adaptive TTLSA models (blue curves), corresponding to TTLSA with 64 or 512 samples and the oracle method, all show a U-shaped curve, which is an upper bound on all the other curves. In particular, we see that the U-shaped curve is tangent to the invariant line when the target distribution is unconfounded ($\lambda = 0.5$); in that case, there is no prior information from $p_t(y, z)$ to exploit. However, in all other target domains, the adapted prior information improves performance. As we increase the size of the unlabeled target dataset, we see that the performance of TTLSA approaches that of the oracle. To illustrate the fact that our method can also be applied to non-neural net classifiers we also applied TTLSA on top a gradient boosted tree classifier. The results (shown in the supplementary) are qualitatively similar to those in Figure 2a.

## 4.2   CheXpert

Next, we apply our method to the problem of disease classification using chest X-rays based on the CheXpert [Irvin et al., 2019] dataset. Chest X-rays are a particularly relevant application area for our method because sensitive attributes, such as patient sex or self-reported race, can be readily predicted from chest X-rays [Gichoya et al., 2022]. Recent work, e.g., Jabbour et al. [2020] and Makar et al. [2022], has confirmed the potential for classifiers to exploit these features as spurious features.

**Dataset**  CheXpert has 224,316 chest radiographs of 65,240 patients. See the supplementary material for a visualizaiton of this dataset. Each image is associated with 14 disease labels derived

from radiology reports, and 3 potentially confounding attributes (age, sex, and race), as listed in the supplementary. We binarized the attributes as in Jabbour et al. [2020], taking age to be 0 if below the median and 1 if above, and sex to be 0 if female and 1 if male. As for the class labels, we define class 0 as "negative" (corresponding to no evidence of a disease), and class 1 as "positive" (representing the presence of a disease); images labeled "uncertain" for the attribute of interest are discarded. Following [Glocker et al., 2022], we focus on predicting the label $y$ = "Pleural Effusion" and use sex as the confounding variable $z$. See Figure 4 for some samples from the dataset. For the input features $\boldsymbol{x}$, we either work with the raw gray-scale images, rescaled to size $224 \times 224$, or we work with 1376-dim feature embeddings derived from the pretrained CXR model [Sellergren et al., 2022]. This embedding model was pre-trained on a large set of weakly labeled X-rays from the USA and India. Note, however, that the pre-training dataset for CXR is distinct from the CheXpert dataset we use in our experiments.

To compare performance under distribution shift, we created a set of 21 distributions, $p_\lambda = (1 - \lambda)p_0 + \lambda p_1$, where $\lambda \in \{0, 0.05, \ldots, 0.95, 1.0\}$ as before. We train using $\lambda = 0.05$, representing a strong spurious correlation at training time, and test with all 21 values. The test images are distinct from the training, and each test distribution has 512 samples in total. Each patient may have multiple images associated with them, but there is no patient overlap in the training and test distributions.

**Training procedure** When working with embeddings, we use a linear logistic regression model, following Sellergren et al. [2022], due to its simplicity and its good performance. To train this, we use AdamW with a batch size of 64 and a learning rate of $10^{-3}$ for a maximum of 5000 epochs, and run calibration for a maximum of 1000 epochs with the same learning rate and batch size on a 10% holdout set. However, training typically terminates much earlier since we also calculate the exponential moving average of the validation loss with a decay rate of 0.1, and stop early when the smoothed validation loss does not decrease for 5 consecutive epochs. When working with pixels, we use a CNN. See the supplementary for details.

**Results** Our results for the embedding version of the data are shown in Figure 2b. The trends are essentially the same as in the colored MNIST case. In particular, we see that our TTLSA method outperforms the invariant baselines, and both adaptive and invariant methods outperform ERM. These results for the pixel version of the data are shown in Appendix C.3 in the supplementary. We find that the relative performance of the methods is similar to the embedding case, but the absolute performance for all methods is better when using embeddings, as was previously shown in Sellergren et al. [2022].

**Discussion** Glocker et al. [2022] point out that embeddings derived from X-ray classification models may contain information about sensitive attributes $z$, such as sex and age. We confirmed this result, and were able to classify sex with an accuracy of over 95% just using logistic regression on the CXR embeddings, as shown in Table 3. Glocker et al. [2022] argue that this may be harmful, since it can cause bias in the predictions of the primary label $y$ of interest (disease status). As a counterpoint, our results also show that, if we can predict both $z$ and $y$ from the embeddings, this information can be used to make optimal adjustments for target populations featuring very different demographic makeups, in ways that can be beneficial for all groups. However, our results also confirm such information does need to be handled with care.

### 4.3 Worst-group vision and text benchmark datasets

In this section, we apply our method to four benchmark datasets that were first introduced in the group DRO paper [Sagawa et al., 2020], and have since been widely used in the literature on group robustness [see, e.g. Idrissi et al., 2022, on data balancing]. The four datasets are as follows.

**CelebA** Introduced in [Liu et al., 2015], this is an image dataset of celebrity faces. The class label $y$ is hair color (blond / not-blond) and the group / attribute label $z$ is sex (male / female).

**Waterbirds** Introduced in [Wah et al., 2011, Sagawa et al., 2020], this is an image dataset of birds synthetically pasted onto two different kinds of backgrounds. The class label $y$ is bird type (land bird or water bird), and the group / attribute label $z$ is background type (land or water).

**MultNLI**  Introduced in [Williams et al., 2018], this is a dataset of sentence pairs, $(s_1, s_2)$, where the goal is to predict if $s_1$ entails $s_2$. The class label $y$ corresponds to entailment, contradiction or neutral, and the group / attribute label $z$ indicates presence / absence of negation words.

**CivilComments (CC)**  Introduced in [Borkan et al., 2019], This is a dataset of sentences from online forums. The class label $y$ represents if the comment is toxic or not, and the group/attribute label $z$ represents whether the content is related to a minority group (such as LGBT) or not.

**Training procedure**  We use the code and hyper-parameters specified in Idrissi et al. [2022] for the ERM, gDRO, and SUBG baselines. We also use these same hyper-parameter values as ERM when fitting our own model TTLSA, which is trained to predict the augmented labels $m = (y, z)$ using logit adjustment. We run the experiments on sixteen Nvidia A100 40GB GPUs. Depending on the dataset and method used, each experiment takes somewhere between 1 hour and 10 hours on an A100. In total, the experiments took 415 GPU hours.

For each method, for worst-group accuracy evaluations, we tune the model using worst-group accuracy in a validation set, whereas for average accuracy evaluations, we tune the model using average validation set accuracy. For worst-group evaluations, we use the LA baseline, without adjusting to the test set distribution. This effectively targets a balanced group distribution, similar to SUBG. For average target accuracy evaluations, we use the full TTLSA method with test-time EM adjustment.

**Results**  We summarize our results in Table 2. We report the worst group and average group accuracy, averaged across 4 replication runs. Our worst group numbers for the baseline methods are within error bars for those reported in Idrissi et al. [2022].[4] The Idrissi et al. [2022] paper does not report average group performance, but we computed these results by modifying their code.

| Data | $m_s$ | $m_t$ | ERM | gDRO | SUBG | LA/TTLSA |
|---|---|---|---|---|---|---|
| CelebA | 0.44 | 0.49 | 80.83 (1.46) / **95.93** (0.03) | **87.36** (0.47) / 94.68 (0.07) | **87.10** (1.26) / 93.44 (0.19) | 84.72 (0.58) / **95.55** (0.09) |
| Waterbirds | 0.73 | 0.39 | 85.78 (0.24) / 93.19 (0.16) | 87.98 (0.86) / 93.06 (0.62) | **88.87** (0.14) / 93.48 (0.11) | **88.38** (0.36) / **95.23** (0.34) |
| MultiNLI | 0.49 | 0.49 | 68.60 (0.40) / **82.70** (0.02) | **76.79** (1.24) / 81.16 (0.07) | 67.89 (0.91) / 72.15 (0.25) | **76.33** (1.45) / **82.60** (0.04) |
| CC | 0.55 | 0.65 | 68.16 (1.03) / **88.00** (0.03) | **79.66** (0.17) / 84.46 (0.43) | 76.56 (0.25) / 79.56 (0.77) | **79.27** (1.17) / 85.03 (0.71) |

Table 2: Accuracy of the worst / average $(y, z)$ group on the benchmark datasets. We define $m_s = \max(\pi_s)$ and $m_t = \max(\pi_t)$ as the maximum probability of a $(y, z)$ group in the source and target distributions. The difference between these values reflects the degree of distribution shift.

Overall, the results show that TTLSA offers a unified approach to achieve a variety of robustness objectives. In terms of average group accuracy, we find that the performance of TTLSA relative to ERM depends on the nature of the shift. For CelebA and MultiNLI, there is no significant distribution shift, so TTLSA is similar to ERM, as expected. For Waterbirds, the majority class in the source becomes much less common in the target (reflecting the fact that the rare combinations of water-birds on land and land-birds on water become more frequent). TTLSA is able to adapt to this, and outperforms ERM. For CivilComments, the majority class becomes even more common in the target distribution. Although TTLSA can adapt to this change, it cannot match the fact that ERM has been "rewarded" for learning a representation that is optimized for a single majority class. In terms of worst-group accuracy, the unadapted LA baseline (which is a special case of TTLSA where we do not use EM adaptation) achieves competitive performance to the dedicated gDRO and SUBG methods.

## 5   Conclusions and future work

We have shown that adapting to changes in the nuisance factors $Z$ can give better results than using classifiers that are designed to be invariant to such changes. However, a central weakness of our approach is that it requires that the generative distribution $p(\boldsymbol{x}|y, z)$ be preserved across domains. This assumption becomes more plausible the more factors of variation are included in $Z$. However, as $Z$ becomes high dimensional, each step of our TTLSA method, as well as access to appropriate

---

[4]Our worst group accuracy results for CivilComments are very different from those reported in Idrissi et al. [2022] despite using their code. The reason is that they binarize the $z$ label during training, but use the fine-grained dataset with 9 possible $z$ values during evaluation. Our method requires that $z$ have the same set of possible values in train and test, so we use the coarse-grained dataset for both training and evaluation, and thus our performance metric is incomparable to theirs.

data, becomes more challenging. In this vein, another weakness is that we require access to labeled examples of $Z$ during training. In the future, we would like to relax this assumption, potentially by using semi-supervised methods [c.f., Sohoni et al., 2021, Lokhande et al., 2022, Nam et al., 2022] that combine small fully labeled datasets with large partially labeled datasets.[5] We also plan to explore the use of fully unsupervised estimates of the confounding factors $Z$, based on generative models, or by leveraging multiple source domains, similar to [Jiang and Veitch, 2022].

We discuss potential societal impacts in the appendix.

## Acknowledgements

We would like to thank Andrew Sellergren at Google Health for his help with the CheXpert CXR embeddings, and Jonathan Caton at Google TPU Research Cloud for providing computational resources.

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

## A   Derivation of the EM algorithm

In this section we describe how to estimate the label prior on the target distribution, $p_t(y, z) = p_t(m) = \pi_m$, using the unlabeled data $\mathcal{D}_t^x$. There are several approaches to this, including a moment matching method called black box shift learning [Lipton et al., 2018] and an MLE approach based on the EM algorithm [Saerens et al., 2002]. In [Alexandari et al., 2020], they show that the MLE approach is much better, provided the classifier is calibrated. (See also [Garg et al., 2020] for a unified analysis of these two approaches.)

Since our augmented label space is expanded to include both class labels $y$ and meta-data $z$, the number of labels $M$ can be large, which can result in problems when computing the MLE. We therefore expand the previous approach to compute the MAP estimate, using a Dirichlet prior of the form

$$\mathrm{Dir}(\boldsymbol{\pi}|\boldsymbol{\alpha}) = \frac{1}{B(\boldsymbol{\alpha})} \prod_{m=1}^{M} \boldsymbol{\pi}_m^{\alpha_m - 1} \tag{9}$$

where $B(\boldsymbol{\alpha})$ is the normalization constant. Note that the MLE solution can be recovered by setting $\boldsymbol{\alpha} = \mathbf{1}$, which represents a uniform prior.

The goal is to maximize the (unnormalized) log posterior of $\boldsymbol{\pi}$ given the unlabeled target data $\mathbf{X}$:

$$\mathcal{L}(\mathbf{X}; \boldsymbol{\pi}) = \log p_t(\boldsymbol{\pi}, \mathbf{X}) \tag{10}$$

$$= \log p_t(\mathbf{X}|\boldsymbol{\pi}) + \log \mathrm{Dir}(\boldsymbol{\pi}|\boldsymbol{\alpha}) \tag{11}$$

$$= \sum_{n=1}^{N} \log p_t(\boldsymbol{x}_n|\boldsymbol{\pi}) + \log \mathrm{Dir}(\boldsymbol{\pi}|\boldsymbol{\alpha}) \tag{12}$$

$$= \sum_{n=1}^{N} \log \left[ \sum_{m=1}^{M} \boldsymbol{\pi}_m p_t(\boldsymbol{x}_n|m) \right] + \log \mathrm{Dir}(\boldsymbol{\pi}|\boldsymbol{\alpha}) \tag{13}$$

The first term can be rewritten as

$$\sum_n \log \left[ \sum_{m=1}^{M} \boldsymbol{\pi}_m p_s(\boldsymbol{x}_n|m) \right] = \sum_n \log \left[ \sum_{m=1}^{M} \boldsymbol{\pi}_m \frac{p_s(m|\boldsymbol{x}_n) p_s(\boldsymbol{x}_n)}{p_s(m)} \right] \tag{14}$$

$$= \sum_n \log \sum_m \frac{p_s(m|\boldsymbol{x})}{p_s(m)} \boldsymbol{\pi}_m + \mathrm{const} \tag{15}$$

This objective is a sum of logs of a linear function of $\boldsymbol{\pi}$, as is the log prior. This needs to be maximized subject to the affine constraints $\boldsymbol{\pi}_m \geq 0$ and $\sum_{m=1}^{M} \boldsymbol{\pi}_m = 1$, so the problem is concave, with a unique global optimum [Alexandari et al., 2020].

One way to compute this optimum is to use EM. Let $\boldsymbol{\pi}^j$ be the estimate of $\boldsymbol{\pi}$ at iteration $j$; we initialize with $\boldsymbol{\pi}_m^0 = p_s(m)$. First note that

$$p_t(\boldsymbol{x}_n, m_n) = p_s(\boldsymbol{x}_n|m_n) p_t(m_n) = \prod_{m=1}^{M} [p_s(\boldsymbol{x}_n|m) \boldsymbol{\pi}(m)]^{\mathbb{I}(m_n = m)} \tag{16}$$

Hence the complete data log posterior is given by

$$\mathcal{L}(\mathbf{X}, \mathbf{M}; \boldsymbol{\pi}) = \sum_{n=1}^{N} \sum_{m=1}^{M} \mathbb{I}(m_n = m) \log[\boldsymbol{\pi}_m p_s(\boldsymbol{x}_n|m)] + \log \mathrm{Dir}(\boldsymbol{\pi}|\boldsymbol{\alpha}) \tag{17}$$

so the expected complete data log posterior is

$$Q\left(\boldsymbol{\pi}, \boldsymbol{\pi}^{(j)}\right) = E_{\mathbf{M}}[\mathcal{L}(\mathbf{X}, \mathbf{M}; \boldsymbol{\pi})|\mathbf{X}, \boldsymbol{\pi}^{(j)}] \tag{18}$$

$$= \sum_{n=1}^{N} \sum_{m=1}^{M} p(m_n = m|\mathbf{X}, \boldsymbol{\pi}^j) \log(\boldsymbol{\pi}_m p_s(\boldsymbol{x}_n|m)) + \log \mathrm{Dir}(\boldsymbol{\pi}|\boldsymbol{\alpha}) \tag{19}$$

$$= \sum_{m=1}^{M} N_m^j \log(\boldsymbol{\pi}_m p_s(\boldsymbol{x}_n|m)) + \sum_{m=1}^{M} (\alpha_m - 1) \log \boldsymbol{\pi}_m - \log B(\boldsymbol{\alpha}) \tag{20}$$

$$= \sum_{m=1}^{M} N_m^j \log \boldsymbol{\pi}_m + \underbrace{\sum_{m=1}^{M} N_m^j \log p_s(\boldsymbol{x}_n|m)}_{\mathrm{const}} + \sum_{m=1}^{M} (\alpha_m - 1) \log \boldsymbol{\pi}_m + \mathrm{const} \tag{21}$$

where we drop constants wrt $\boldsymbol{\pi}$, and where we defined the expected counts to be

$$N_m^j = \sum_{n=1}^{N} p(m_n = m|\boldsymbol{x}_n, \boldsymbol{\pi}^j) \tag{22}$$

Hence in the E step we just need to compute the posterior responsibilities for each label:

$$p(m_n = m|\boldsymbol{x}_n, \boldsymbol{\pi}^j) = \frac{\boldsymbol{\pi}^j(m) p_s(\boldsymbol{x}_n|m)}{\sum_{m'=1}^{M} \boldsymbol{\pi}^j(m') p_s(\boldsymbol{x}_n|m')} = \frac{\boldsymbol{\pi}^j(m) p_s(m|\boldsymbol{x}_n)/p_s(m)}{\sum_{m'=1}^{M} \boldsymbol{\pi}^j(m') p_s(m'|\boldsymbol{x}_n)/p_s(m')} \tag{23}$$

We plug this into Equation (22) and then maximize Equation (21), using a Lagrange multiplier to enforce the sum to one constraint. We then get the following (see e.g., Sec 4.2.4 of Murphy [2022] for the derivation):

$$\hat{\boldsymbol{\pi}}_m^{j+1} = \frac{\tilde{N}_m^j}{\sum_{m'=1}^{M} \tilde{N}_{m'}^j} \tag{24}$$

where $\tilde{N}_m^j$ are the prior pseudo counts plus the expected empirical counts:

$$\tilde{N}_m^j = N_m^j + \alpha_m - 1 \tag{25}$$

At convergence, we have

$$p_t(y, z) = \hat{\boldsymbol{\pi}}_{y,z}^J \tag{26}$$

If we assume that the class label prior is constant, and only the distribution of auxiliary labels has changed, then we can write

$$p_t(y, z) = p_s(y) p_t(z|y) \tag{27}$$

where

$$p_t(z|y) = \frac{p_t(y, z)}{\sum_{z'} p_t(y, z')} \tag{28}$$

However, we do not make this fixed label assumption in our experiments.

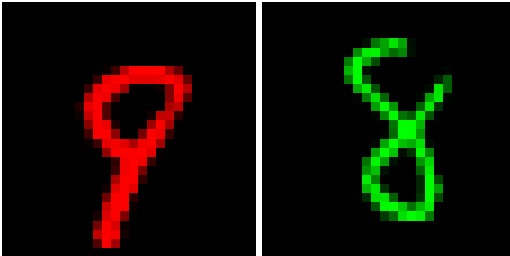

Figure 3: Samples from ColoredMNIST. (a): $y = 1$, $z = 0$. (b) $y = 1$, $z = 1$.

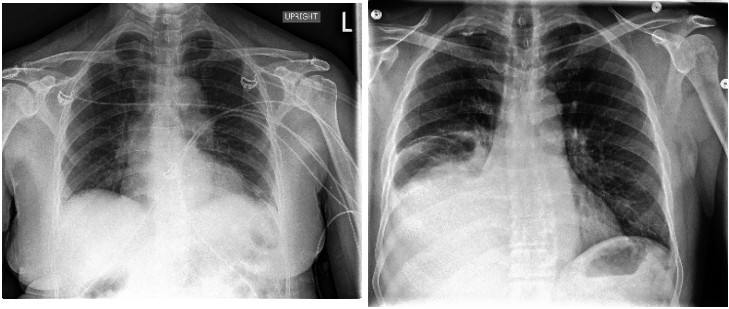

Figure 4: Samples from CheXpert. **Left**: Female patient without effusion. **Right**: Male patient with effusion.

## B   Datasets

In this section we discuss the datasets in more detail.

### B.1   Colored MNIST

We show some sample images in Figure 3.

### B.2   CheXpert

We show some sample images in Figure 4. We list all the target attributes in Table 3. To test the difficult of each task, we train a logistic regression model for each attribute on the embeddins. (We get similar results using an MLP.) The resulting AUC scores are shown in Table 3. This shows we can reliably predict all the attributes from the embeddings. The table also shows the marginal distribution of each attribute. Many labels are highly skewed, which means accuracy would be a poor measure of the predictive performance.

Interestingly, we see that we can predict sex with an AUC of 0.973, which is higher than the AUC for effusion (0.861). To understand why, note that we only use frontal scans; consequently breasts are often visible in female patients, and this is often easier to detect visually than detecting the disease itself (see Figure 4), providing a possible "shortcut" for models to exploit.

| Attribute | AUC | Prob. |
|---|---|---|
| NO_FINDING | 0.873 | 0.909 |
| ENLARGED_CARDIOMEDIASTINUM | 0.652 | 0.942 |
| CARDIOMEGALY | 0.843 | 0.867 |
| AIRSPACE_OPACITY | 0.711 | 0.480 |
| LUNG_LESION | 0.761 | 0.963 |
| PULMONARY_EDEMA | 0.848 | 0.696 |
| CONSOLIDATION | 0.683 | 0.911 |
| PNEUMONIA | 0.742 | 0.973 |
| ATELECTASIS | 0.694 | 0.815 |

| Attribute | AUC | Prob. |
|---|---|---|
| PNEUMOTHORAX | 0.883 | 0.875 |
| EFFUSION | 0.861 | 0.508 |
| PLEURAL_OTHER | 0.752 | 0.987 |
| FRACTURE | 0.784 | 0.962 |
| SUPPORT_DEVICES | 0.900 | 0.420 |
| GENDER | 0.973 | 0.586 |
| AGE_AT_CXR | 0.914 | 0.492 |
| PRIMARY_RACE | 0.731 | 0.459 |
| ETHNICITY | 0.681 | 0.728 |

Table 3: Metrics for all the attributes in the CheXpert dataset. (a) AUC using Logistic Regression on CXR embeddings. (b) Baseline prior probability for each attribute, illustrating the severe class imbalance for many attributes.

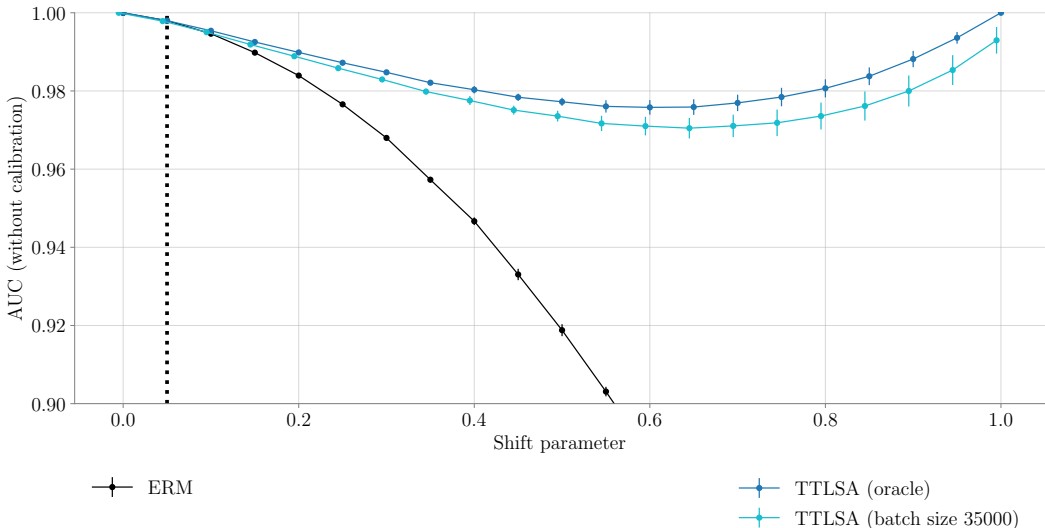

Figure 5: Performance on Colored MNIST using an uncalibrated tree classifier. TTSLA still improves the performance of the base model.

## C  Extra results

In this section, we include some extra experimental results.

### C.1  Colored MNIST using gradient boosted tree classifier

In Figure 5, we show the results of various methods on the Colored MNIST dataset, where we use a Gradient Boosting Classification Tree as our base classifier, instead of a DNN. In particular, we use the `HistGradientBoostingClassifier` from scikit-learn [Pedregosa et al., 2011] with default parameters. The results are qualitatively similar to the DNN case.

### C.2  The benefits of calibration

In Figure 6 we show the results on CheXpert if we remove the calibration step for our base classifier. Compared to Figure 2b, we see that the overall AUC of all the methods is worse, and the variance is larger. [6] However, the rank ordering of the methods is the same. It is notable that a large gap opens up between the Oracle curve and the TTLSA implementations. This suggests that calibration primarily improves estimation of $p_t(y, z)$ estimation via EM, because the Oracle curve in this subfigure corresponds to using the correct weights with the uncalibrated $p_s(y, z|\boldsymbol{x})$ model.

### C.3  CheXpert using CNN on raw pixels

In Figure 7 we show the result of various methods when applied to CheXpert images, as opposed to using embeddings. We use a ResNet-50 that was pretrained on Imagenet, which we then fine tune on CheXpert images by replicating the gray-scale image along all 3 RGB channels. The qualitative conclusions are the same as in the embedding case.

### C.4  More results on the benchmark datasets

In Table 4 and Table 5 we report the per-group accuracy on the benchmark datasets.

---

[6]For a binary classification problem, calibration will not change the AUC, but since we derive the posterior over class labels by marginalizing a 4-way joint, $p(y|\boldsymbol{x}) = \sum_{z=0}^{1} p(y, z|\boldsymbol{x})$, calibration can help.

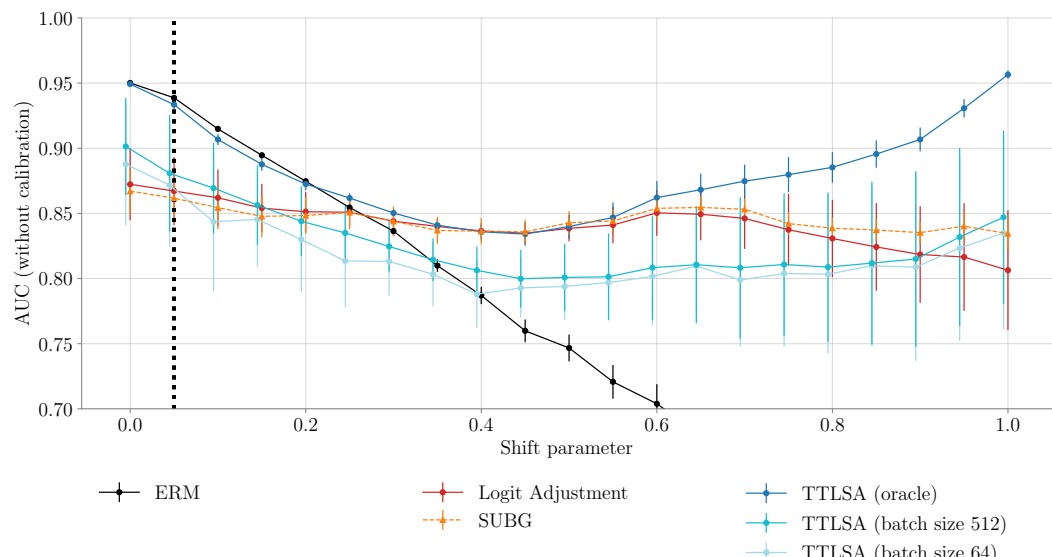

Figure 6: Performance across target domains on CheXpert embeddings, following the setup of Figure 2a. (a) Results using calibration. Performance mirrors those in Figure 2a. (b) Results without calibration. We see that calibration both improves performance and decreases variability between runs.

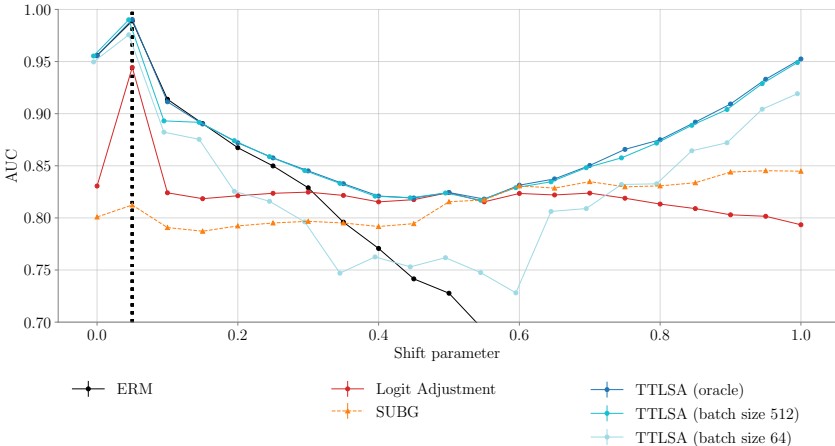

Figure 7: Performance on CheXpert using raw image (pixel) input instead of embeddings. These results are with calibration.

| Data | Method | Group label $(Y, Z)$ | | | | | |
|---|---|---|---|---|---|---|---|
| | | (0, 0) | (0, 1) | (1, 0) | (1, 1) | (2, 0) | (2, 1) |
| CelebA | ERM | 86.71 (0.67) | 92.65 (0.79) | 96.80 (0.18) | 80.83 (1.46) | | |
| | gDRO | 92.71 (0.05) | 92.33 (0.09) | 92.63 (0.35) | 87.36 (0.47) | | |
| | SUBG | 91.76 (0.22) | 91.91 (0.55) | 90.96 (0.33) | 87.22 (1.38) | | |
| | LA | 91.43 (0.11) | 94.77 (0.08) | 95.88 (0.12) | 84.72 (0.58) | | |
| | TTLSA | 97.59 (0.18) | 98.78 (0.03) | 80.70 (1.22) | 51.25 (0.27) | | |
| Waterbirds | ERM | 98.95 (0.13) | 86.57 (0.48) | 86.02 (0.13) | 95.76 (0.16) | | |
| | gDRO | 93.46 (0.22) | 88.00 (0.88) | 90.15 (0.10) | 92.06 (0.34) | | |
| | SUBG | 90.76 (0.69) | 88.96 (0.19) | 91.28 (0.35) | 91.36 (0.24) | | |
| | LA | 94.43 (1.63) | 88.38 (0.36) | 91.32 (0.43) | 93.15 (0.54) | | |
| | TTLSA | 94.59 (0.44) | 93.68 (0.73) | 95.72 (0.29) | 97.12 (0.19) | | |
| MultiNLI | ERM | 80.75 (0.79) | 94.94 (0.11) | 83.18 (0.47) | 78.05 (1.31) | 81.98 (0.48) | 68.60 (0.40) |
| | gDRO | 80.36 (0.63) | 85.27 (0.25) | 82.48 (0.59) | 81.21 (1.30) | 79.39 (1.34) | 76.87 (1.28) |
| | SUBG | 69.63 (0.17) | 82.85 (0.19) | 74.39 (0.21) | 79.68 (0.17) | 69.84 (0.48) | 68.40 (1.33) |
| | LA | 81.63 (1.15) | 87.79 (1.99) | 84.36 (0.89) | 80.95 (2.03) | 78.77 (1.09) | 76.33 (1.45) |
| | TTLSA | 80.24 (0.87) | 94.74 (0.58) | 81.73 (2.45) | 73.90 (1.72) | 82.40 (1.49) | 63.76 (2.15) |
| CivilComments | ERM | 92.23 (0.42) | 90.38 (0.46) | 68.57 (1.07) | 68.32 (0.97) | | |
| | gDRO | 83.94 (0.70) | 79.92 (0.33) | 80.97 (0.63) | 81.09 (0.42) | | |
| | SUBG | 79.79 (0.56) | 79.14 (0.34) | 82.52 (0.46) | 76.56 (0.25) | | |
| | LA | 84.45 (0.16) | 79.27 (1.17) | 83.00 (0.95) | 84.20 (0.99) | | |
| | TTLSA | 85.53 (1.35) | 74.94 (1.96) | 84.34 (2.36) | 84.61 (2.21) | | |

Table 4: Per-group accuracy on the benchmark datasets, where model selection is based on the worst $(Y, Z)$ group accuracy on a validation set. Numbers in parentheses signify the standard error calculated based on 4 replication runs.

| Data | Method | Group label $(Y, Z)$ | | | | | |
|---|---|---|---|---|---|---|---|
| | | (0, 0) | (0, 1) | (1, 0) | (1, 1) | (2, 0) | (2, 1) |
| CelebA | ERM | 96.54 (0.21) | 99.58 (0.05) | 86.47 (0.89) | 40.28 (2.24) | | |
| | gDRO | 95.48 (0.14) | 96.76 (0.14) | 87.10 (0.34) | 68.75 (1.14) | | |
| | SUBG | 95.45 (0.39) | 95.93 (0.44) | 79.81 (1.30) | 67.64 (4.66) | | |
| | LA | 95.25 (0.55) | 98.96 (0.49) | 88.77 (1.81) | 43.47 (12.05) | | |
| | TTLSA | 97.68 (0.13) | 98.99 (0.07) | 80.21 (1.04) | 47.36 (1.81) | | |
| Waterbirds | ERM | 99.42 (0.11) | 90.27 (1.24) | 80.61 (2.51) | 94.16 (0.84) | | |
| | gDRO | 97.04 (1.44) | 92.84 (1.07) | 83.84 (2.47) | 89.10 (0.82) | | |
| | SUBG | 96.98 (0.29) | 95.88 (0.42) | 82.87 (1.51) | 83.33 (1.86) | | |
| | LA | 98.23 (0.15) | 92.42 (0.36) | 85.98 (1.06) | 92.91 (0.19) | | |
| | TTLSA | 99.06 (0.11) | 93.61 (1.04) | 87.66 (0.40) | 95.02 (0.92) | | |
| MultiNLI | ERM | 82.43 (0.06) | 95.47 (0.08) | 83.62 (0.03) | 77.14 (0.16) | 80.45 (0.09) | 67.36 (0.54) |
| | gDRO | 80.37 (0.82) | 86.32 (0.64) | 81.06 (0.72) | 78.22 (0.60) | 81.22 (0.22) | 78.83 (0.29) |
| | SUBG | 68.30 (2.00) | 83.95 (2.28) | 75.72 (1.68) | 79.40 (1.37) | 69.91 (1.55) | 66.44 (2.23) |
| | LA | 82.74 (0.06) | 92.92 (0.34) | 83.97 (0.44) | 79.88 (0.55) | 79.77 (0.34) | 71.49 (0.95) |
| | TTLSA | 81.86 (0.19) | 96.52 (0.11) | 83.89 (0.37) | 76.07 (0.71) | 80.51 (0.17) | 56.60 (1.36) |
| CivilComments | ERM | 96.00 (0.38) | 95.63 (0.53) | 55.27 (1.88) | 52.21 (2.43) | | |
| | gDRO | 89.59 (0.68) | 86.60 (0.86) | 71.56 (1.64) | 71.94 (1.33) | | |
| | SUBG | 81.43 (1.09) | 80.67 (1.28) | 80.80 (1.34) | 76.05 (0.44) | | |
| | LA | 84.45 (0.16) | 79.27 (1.17) | 83.00 (0.95) | 84.20 (0.99) | | |
| | TTLSA | 91.53 (0.58) | 87.18 (1.70) | 70.14 (2.69) | 71.32 (2.82) | | |

Table 5: Per-group accuracy on the benchmark datasets, where model selection is based on the average $(Y, Z)$ group accuracy on a validation set. Numbers in parentheses signify the standard error calculated based on 4 replication runs.

## C.5 Training with partial group labels

In this section, we evaluate an extension of our method where not all training samples have group labels $z$. In particular, we first train an ERM model to predict $z$ on samples with group labels $z$, calculate $p(z|x)$ for training samples with missing $z$, and then fit a new $p(y, z|x)$ model on the augmented data. In particular, we represent each $(y, z)$ target as a one-hot vector when $z$ is known, and use a soft (predicted) encoding when $z$ is unknown. We train with cross entropy loss. The use of soft labels may have the benefits of self-distillation Pham et al. [2022]. The validation set is always fully labeled for the purpose of hyperparameter tuning.

The results (on the 4 benchmark datasets) are shown in Table 6. The accuracy barely drops as missingness increases, which means our method is robust to the deficiency in group labels $z$.

| Data | Missingness | | | | |
| --- | --- | --- | --- | --- | --- |
| | 0 | 0.5 | 0.75 | 0.875 | 0.9375 |
| CelebA | 84.72 / 95.55 | 78.33 / 95.68 | 77.78 / 95.37 | 79.44 / 94.44 | 77.22 / 95.20 |
| Waterbirds | 88.38 / 95.23 | 87.63 / 93.98 | 88.79 / 94.41 | 88.65 / 94.67 | 91.28 / 95.05 |
| MultiNLI | 76.33 / 82.60 | 74.87 / 79.55 | 74.72 / 82.49 | 76.05 / 82.61 | 78.75 / 81.72 |
| CivilComments | 79.27 / 85.03 | 76.26 / 85.87 | 73.87 / 83.55 | 73.41 / 84.57 | 66.64 / 80.36 |

Table 6: Accuracy of the worst / average $(y, z)$ group on the benchmark datasets with partial training $z$ labels, where model selection is based on average $z$ accuracy. The *Missingness* columns stand for the proportion of training set with missing labels, e.g. 0.75 means only 25% of the training samples have $z$ labels.

# D   Potential negative societal impacts

The proposed method in this work yields a model that can adapt to a new distribution and improves the performance at test time by exploiting spurious correlations to create a label shift correction technique that adapts to changes in the marginal distribution $p(y, z)$ using unlabeled samples from the target domain. In this way, there are potential societal benefits to our method, especially when $z$ corresponds to a socially salient attribute, such as a protected class. However, use cases of this type require caution, especially given the limitations discussed in Section 5. Further, as we discuss in a footnote in the main text, our method does not address concerns about cases where making decisions on the basis of $z$ is discouraged or forbidden for *a priori* reasons. Given these limitations, there is a potential that the existence of adaptation methods of this type could be used to downplay the potential dangers of misusing sensitive information in machine learning systems. Here, we hope researchers and practitioners will instead acknowledge that, while beneficial use cases of $z$ information exist, (1) there is a need to validate empirically that a particular use of $z$ information is actually socially beneficial, and (2) there are valid reasons why one might want to avoid using $z$ information altogether. Further, there is a potential risk that if the measurement quality of the labels $y, z$ shift across distributions, such that they measure distinct concepts, or exhibit substantially different noise properties (i.e., become biased, or exhibit more outliers), our framework might absorb them during adaptation and eventually the outcomes of the system might be biased as well.

# E  Invariance Equivalences and Conditions

In this section, we review connections that have been established between risk invariance, ERM on balanced data, "separation" between a predictor $f(X)$ and the spurious factor $Z$, and worst-$(y, z)$-group performance. These results are useful for understanding why the application of logit adjustment at training time often yields a predictor that exhibits approximately invariant risk across the test sets that we study in our experiments.

## E.1  Key Concepts

**Risk invariance**   A predictor is risk-invariant with respect to a loss function $\ell$ and a family of test distributions $\mathcal{Q}$ iff it has the same risk $E_Q[\ell(f(X), Y)]$ for each $Q \in \mathcal{Q}$. The results we discuss apply to test distribution families that preserve both the generative distribution *and* the label distribution of the source distribution; that is, $\mathcal{Q}$ is the set of distributions such that $Q(Y) = P(Y)$ and $Q(X \mid Y, Z) = P(X \mid Y, Z)$ for each $Q \in \mathcal{Q}$. This formulation allows $Q(Z \mid Y)$ can change. This is is the family is considered in Makar et al. [2022] and Makar and D'Amour [2022], and is called a "causally compatible" family in Veitch et al. [2021], or a correlation shift in Yi et al..

**Pure spuriousness**   The data generating process in Figure 1 is purely spurious if there exists some sufficient statistic $e(X)$ such that (1) $Y \perp X \mid e(X)$ and (2) $e(X) \perp Z \mid Y$. In words, if we know $e(X)$, there is no further dependence between $Y$ and $X$, and further, $e(X)$ does not depend on the spurious factor $Z$ except through $Z$'s marginal dependence with $Y$. This is consistent with a causal model where the influence of $Y$ on $X$ is totally mediated by $e(X)$, and $Z$ has no causal effect on $e(X)$.

Veitch et al. [2021] coined the term "purely spurious" in a context of a full counterfactual model of data generation, to refer to data generating processes where the portions of $X$ that are causally related to $Y$ and $Z$ can be separated in a specific sense. Makar et al. [2022] consider the special case of pure spuriousness in the context of the anti-causal model in Figure 1. (They do not use the term "purely spurious" as the work in Veitch et al. [2021] was concurrent; Makar and D'Amour [2022] makes the connection explicit.) Here, we use formalism from Makar et al. [2022] to present the idea to minimize conceptual overhead.

Note that when the data $X$ is rich, such as images are long passes of text, pure spuriousness is more plausible (or a better approximation to reality) because there is less possibility of descructive interfecence between $Y$ and $Z$ in the generation of $X$. Specifically, the simplest examples where pure spuriousness fails are ones where $X$ is very low-content: e.g., $Y$ and $Z$ are binary, and $X := Y$ OR $Z$.

**Separation**   Separation is a concept popularized in the literature on ML fairness [Barocas et al., 2019, Chapter 3], which stipulates that the predictor $f(X)$ should satisfy the the conditional independence $f(X)Z \perp \mid Y$. When $Z$ is a sensitive attribute, this condition stipulates that the predictor $f(X)$ should contain no more information about $Z$ than one could glean from knowing $Y$ alone.

**Data balancing**   Idrissi et al. [2022] study predictors trained on data subsampled so that the $(Y, Z)$ distribution is uniform; they call this data-balancing. Makar et al. [2022] and Makar and D'Amour [2022] study a similar predictors optimized on a similar "ideal" distribution, where $Q(Y, Z) = P(Y)P(Z)$ for some source distribution $P$. This distribution does not "balance" the marginals of $Y$ and $Z$, but it eliminates the marginal correlation between $Y$ and $Z$.

**Worst group performance**   Sagawa et al. [2020] define groups in terms of $(z, y)$ values. The group conditional risk is $R_{z,y} = E_Q[\ell(f(X), Y) \mid Z = z, Y = y]$. Note that for all families of test sets that we consider, the group-conditional risks are equal for all Q. Worst group risk minimization attempts to minimize the group conditional risk of the worst subgroup. Saerens et al. [2002] propose a distributionally robust optimization algorithm for performing this minimization.

## E.2  Connections

In the purely spurious setting, there are several connections and near-equivalences between risk invariance, separation, optimality on balanced data, and worst group risk minimization.

Yi et al. establish that for label distribution preserving target families, a predictor $f(X)$ that satisfies separation $f(X) \perp Z \mid Y$ will have invariant risk across the family $\mathcal{Q}$ defined above. Notably, this result does *not* require pure spuriousness.

Under pure spuriousness, the separation condition achieves a certain optimality. Veitch et al. [2021], Theorem 4.3 establishes that in the purely spurious case, the minimax optimal across the family $\mathcal{Q}$ satisfies separation $f(X) \perp Z \mid Y$. Similarly, under pure spuriousness, Makar and D'Amour [2022], Proposition 2, establishes that the optimal risk-invariant predictor satisfies separation.

Interestingly, this result establishes a connection between optimality under balanced data, separation, and optimal risk invariance. Specifically, Makar et al. [2022], Proposition 1 establishes that the optimal model for the "ideal" uncorrelated distribution for which $Q(Y, Z) = P(Y)P(Z)$ achieves risk invariance across the family $\mathcal{Q}$. Thus, minimizing risk under a separation constraint targets a similar predictor to the predictor that one would target simply optimizing on balanced data. Makar and D'Amour [2022] shows that the near-equivalence holds up empirically, such that learning algorithms targeted at efficiently learning the optimal predictor on balanced data can satisfy both risk invariance and separation criteria.

Idrissi et al. [2022] establish that, empirically, models trained to minimize risk on balanced data also yield favorable worst-group performance, showing that subsamping can be particularly effective. Sagawa et al. [2020] explore similar ideas, focusing on reweighting strategies, which both they and Idrissi et al. [2022] find to work relatively poorly with neural models in the data regimes they study. Sagawa et al. [2020] further establish that under certian convexity conditions, there does exist a reweighting of the data that optimizes worst-group performance, but provide a counterexample showing that this is not always the case with non-convex losses.

Based on the above results, in the purely spurious case, one can establish the following, for $\mathcal{Q}$ with a uniform distribution on $Y$:

1. There exists a predictor $f^*(X)$ that is optimal on the ideal balanced data, is the optimal risk-invariant predictor, and satisfies separation $f(X) \perp Z \mid Y$.

2. For all $Q \in \mathcal{Q}$, the group-specific risks are equal within labels, i.e., $E_Q[\ell(f^*(X), Y) \mid Y = y, Z = z] = E_Q[\ell(f^*(X), Y) \mid Y = y, Z = z']$ for all $y$.

The latter fact does not imply that $f^*(X)$ also optimizes worst-group risk, but it does imply that the worst group cannot be the worst due to a spurious correlation between $Y$ and $Z$. This is because, for a fixed label value $y$, the risks of $(y, z)$ subgroups are the same.

