# OpenReview forum: "Beyond Invariance: Test-Time Label-Shift Adaptation for Addressing "Spurious" Correlations"
_NeurIPS.cc/2023/Conference — NeurIPS 2023 poster_

### Official Review · Reviewer_C4bL · 2023-06-25

**Soundness:** 3 good
**Presentation:** 3 good
**Contribution:** 3 good
**Rating:** 6
**Confidence:** 3

**Summary:**

This paper proposes to perform test-time adaptation to exploit the existence of spurious correlations in a setting where $p(x | y, z)$ remains constant, but when the prior $p(y, z)$ changes based on the target task.

The authors evaluate this new method on various tasks, including standard image/text benchmarks and a medical imaging dataset on chest X-rays. Their empirical results show that their proposed method achieves better performance than standard ERM and the (approximation of a) invariance-based method.


**Strengths:**

The empirical results show that the method achieves better performance than standard ERM and the approximation of an invariant method that assigns a uniform prior over the spurious features on ColoredMNIST and CheXpert.

**Weaknesses:**

Their method relies on a more complex version of the label shift assumption, which must now hold on a much larger set of configurations (|Z| \timex |Y|), which is a weakness (that the authors do indeed point out). However, it’s unclear how reasonable this assumption is to make.

There are no comparisons to other test-time adaptation methods, so the additional amount of unlabeled data may be inflating the benefits in the reported results.

Some small issues with the writing: missing references in the Results and Discussion paragraphs of subsection 4.2, text in figures being difficult to read.

As a side note, I feel that the definition of spurious features used in this paper is a bit counterintuitive. These are useful features for a downstream task, which aren’t what I think of as the standard notion of spurious features.


**Questions:**

In practice, is this stronger label-shift assumption violated? Some additional discussion about this would be appreciated.


**Limitations:**

Limitations are sufficiently discussed.

---

> ### Author Rebuttal · Authors · 2023-08-10
>
> Thank you for your review. Response to your points is below.
>
>
> “**There are no comparisons to other test-time adaptation methods.**”
>
> All the test-time adaptation methods we are aware of are focused on covariate shift, or label shift, but no methods (AFAIK) can handle shift in the correlation between latent factors and the class prior.  Thus these methods are unsuitable for our problem setting.
>
> “**I feel that the definition of spurious features used in this paper is a bit counterintuitive. These are useful features for a downstream task, which aren’t what I think of as the standard notion of spurious features.**”
>
> While it is true that “spurious” suggests the features have no predictive value for the class, in most setups (eg colored MNIST), these features (e.g. background) are in fact predictive of the class, just not in a stable way. Our observation is that we can model this change in correlation, rather than be invariant to it, and thus get better performance.
>
>
> “**In practice, is this stronger label-shift assumption violated?**“
>
> The fact that we get better performance on a variety of datasets is empirical demonstration of the reasonableness of our assumption. However, we do not claim that it captures all (or even most) forms of distribution shift.
>
> “**The method relies on a more complex version of the label shift assumption, which must now hold on a much larger set of configurations ($|Z| \times |Y|)$, which is a weakness (that the authors do indeed point out). However, it’s unclear how reasonable this assumption is to make.**”
>
> This assumption is actually usually weaker than the standard label shift assumption. Specifically, if we can hold more factors of variation fixed, then we can eliminate potential confounding variables that would make the generative distribution of X (p(X | Y) or p(X | Y, Z)) differ between source and target distributions. The tradeoff is that the discriminative model that we need to learn to use our method p(Y, Z | X) may be harder to learn than p(Y | X). However, this is something we can evaluate empirically.

---

> > ### Comment · Reviewer_C4bL · 2023-08-11
> > **Reviewer response**
> >
> > Thanks for the clarifications, especially regarding the weakness of the modified version of the label shift assumption. I'm happy to increase my score from a 5 to a 6, and will watch the responses from other reviewers.

---

### Official Review · Reviewer_1Li8 · 2023-06-29

**Soundness:** 4 excellent
**Presentation:** 4 excellent
**Contribution:** 2 fair
**Rating:** 6
**Confidence:** 4

**Summary:**

This paper introduces a new method named Test-Time Label Shift Adaptation(TTLSA) to manage distribution shifts in machine learning, specifically focusing on spurious correlations.
Rather than eliminating these spurious relations as mentioned in other papers, TTLSA utilizes the spurious features to adjust the model and argues that this could substantially outperform invariant predictors. The method involves augmenting the label space with a "nuisance factor", z, which encapsulates the spurious relations in the data.

TTLSA method involves two main steps:
1. Train on Source: The model is initially trained on the source data with augmented labels that include both the target labels (y) and the nuisance variables (z). This step helps to capture the spurious correlations that exist in the source data.
2. Adapt to Target: During testing, TTLSA uses an EM algorithm to adapt to the target data. This step helps the model to adapt to the shift in label distribution between the training and testing data.

The authors tested TTLSA on several datasets, including the CheXpert and colored MNIST datasets. In both cases, TTLSA outperformed traditional methods in terms of AUC under shifted distributions. However, the authors acknowledge the approach's limitations: it requires the generative distribution to be preserved across domains and needs access to labeled examples of Z during training.

**Strengths:**

**Originality**: The paper introduces a novel method, Test Time Label Shift Adaptation (TTLSA), to handle the problem of distribution shifts in machine learning models. It leverages the spurious relations in the data rather than attempting to eliminate them. This unique approach sets it apart from previous methods.

**Quality**: The paper is of good quality. The authors' method is well-defined, and the experiments are carefully designed, involving various benchmark datasets to demonstrate the effectiveness of the proposed method.

**Clarity**: The paper is well-written and clear, providing a detailed explanation of the methodology, experimental design, and results.

**Significance**: The paper's results on various datasets indicate that TTLSA has the potential to improve machine learning model robustness in the presence of distribution shifts and spurious relations. However, the significance of the method is limited by its assumptions, notably the need for the generative distribution to remain constant across domains. This condition is only based on one additional feature, limiting the method's scope. The results, although promising, should be taken with a degree of caution until further research is conducted to validate these assumptions in more complex scenarios. Overall, the paper holds promise but requires additional exploration and validation.


**Weaknesses:**

1. Assumptions: The TTLSA method assumes that the generative distribution p(x|y, z) is preserved across domains. Here, the authors only address z as a one-dimensional feature. This assumption can be restrictive and may not hold true in many real-world scenarios.

2. Data Requirement: The proposed method requires access to labeled examples of the nuisance factor Z during training. This could be a limitation in scenarios where such labeled data might not be readily available or where it's challenging to identify and label the nuisance factors.


**Questions:**

1. The paper presents great results using the proposed TTLSA method, but the assumption that the generative distribution p(x|y, z) is preserved across domains seems to rest heavily on the presence of one auxiliary feature. Can the authors provide additional insights into scenarios where this assumption is likely to hold? Are there particular types of data or specific domains where this assumption is more valid?

2. Related to the first question, in real-world scenarios, there can be multiple factors affecting the distribution. How would the TTLSA method behave in such complex situations, given its dependency on a single auxiliary feature for its assumptions? Can the authors share their thoughts on extending their methodology to accommodate multiple auxiliary features?

3. The paper mentioned future work that plans to relax the assumption of requiring labeled examples of Z during training. Could the authors provide some preliminary insights into how they plan to achieve this?


**Limitations:**

The authors have very adequately addressed the limitations.

---

> ### Author Rebuttal · Authors · 2023-08-10
>
> Thank you for your thoughtful review. Response to your points is below.
>
> “**The TTLSA method assumes that the generative distribution p(x|y, z) is preserved across domains. Here, the authors only address z as a one-dimensional feature. This assumption can be restrictive and may not hold true in many real-world scenarios.**”
>
> We 100% agree. We leave it to future work to consider multi-variate z. For low-dimensional z, one naive approach is to simply take a cartesian product of its components.
>
> “**The proposed method requires access to labeled examples of the nuisance factor Z during training**”
>
>  We agree this is a weakness. However, as we show in Appendix C.5 (in supplemental), it is possible to apply our method with a very small amount of labeled data. In particular, we get good results on the worst group benchmarks even when up to 90% of the z labels are missing. The trick is to use label imputation to infer the missing z labels before applying our algorithm. We will clarify this in the camera ready.
>
> “**The assumption that the generative distribution p(x|y, z) is preserved across domains seems to rest heavily on the presence of one auxiliary feature. Can the authors provide additional insights into scenarios where this assumption is likely to hold?**“
>
> As mentioned above, we will likely need to make z be a vector of factors (not just a single categorical variable) to make the method more general. We hope that datasets with richer meta-data may provide a source for such structured z’s. Alternatively we may be able to use techniques from the disentangled representation learning literature to avoid the need for extra annotations. However we leave this to future work.
>
> “**Can the authors share their thoughts on extending their methodology to accommodate multiple auxiliary features?**”
>
> We have indeed thought about this. One idea would be to make p(y, z(1:K) | x) be some kind of structured model, such as a (sparse) conditional random field. The key is to capture the dependence between each z(k) and y, conditioned on x. However we leave this to future work.
>
> “**The paper mentioned future work that plans to relax the assumption of requiring labeled examples of Z during training. Could the authors provide some preliminary insights into how they plan to achieve this?**”
>
> See comment above about our experimental results in Appendix C.5 in the supplemental.

---

> > ### Comment · Reviewer_1Li8 · 2023-08-12
> >
> > I will remain the score as 6 but thank you for the clarification

---

### Official Review · Reviewer_wAGR · 2023-07-05

**Soundness:** 2 fair
**Presentation:** 2 fair
**Contribution:** 2 fair
**Rating:** 6
**Confidence:** 4

**Summary:**

**A brief summary:**
The paper proposed "Test-Time Label-Shift Adaptation" (TTLSA), that utilizes rather than eliminates spurious correlations. TTLSA adapts to changes in marginal distribution using the Expectation Maximization algorithm. Experiments on several datasets show that TTLSA outperformed traditional invariance methods and baseline empirical risk minimization.

**Main Contributions:**
The authors proposed a novel test-time adaptation method (TTLSA that capitalizes on spurious correlations instead of trying to eliminate them. Moreover, they expand on the label shift assumption, incorporating nuisance factors into the labels, train a discriminative classifier to predict the source distribution, and adapt to changes in the marginal distribution using the Expectation Maximization (EM) algorithm.

**Strengths:**

Strengths are summarized below:

* The paper introduced an assumption that there exists "a hidden confounder that induces a spurious correlation between the label $Y$ and other causal factors $Z$, which together generate the features $X$," such that the label space in the DRO setting could be extended to $m=(y, z)$, hence treat like classical learning paradigm.

* Experiments on several datasets show that TTLSA achieves promising results compared with traditional invariance methods.

**Weaknesses:**

Main weaknesses are two folds:

* (1) Although the proposed method performs the label shift adaptation at test time, it still relies on a trained discriminative classifier, which is used to predict $p(y, z|x)$ on the source distribution.

* (2) The training on the discriminative classifier requires a large labeled dataset from the source distribution.

**Questions:**

* (1) The introduced model assumption serves as a foundation for the proposed method. Is it possible to empirically verify or illustrate more about the reasoning behind adopting such an assumption?

* (2) It would be better to distinguish the notation of $x_n$, which appears in both the source and target distributions (lines 145-146)

* (3) In Figure 2, can the authors explain more about the $U-shape$ performance of TTLSA? Specifically, why the performance of TTLSA under shift parameter (0.0) might be worse than that of shift parameter (1.0)?

* (4) It seems that certain references to figures are not appropriately included, i.e., line 311.

* (5) It would be better if the authors could discuss the differences with two closely related works, i.e., [1, 2].

References:

[1] Coping with label shift via distributionally robust optimization. [ICLR'21] --> propose an objective to cope with label shift, and provide an adversarial algorithm to effectively optimize it.

[2] Distributionally Robust Post-hoc Classifiers under Prior Shifts. [ICLR'23] --> a method for scaling the model predictions at test-time for improved distribution robustness to prior shifts.

**Limitations:**

Same as mentioned weaknesses.

---

> ### Author Rebuttal · Authors · 2023-08-09
>
> Thank you for your review. Response to your points is below.
>
>
> “**The training on the discriminative classifier requires a large labeled dataset from the source distribution….  still relies on a trained discriminative classifier, which is used to predict p(y, z|x)**”
>
>  We agree this is a weakness. However, as we show in Appendix C.5 (in supplemental), it is possible to apply our method with a very small amount of labeled data. In particular, we get good results on the worst group benchmarks even when up to 90% of the z labels are missing. The trick is to use label imputation to infer the missing z labels before applying our algorithm. We will clarify this in the revised paper.
>
> “**The introduced model assumption serves as a foundation for the proposed method. Is it possible to empirically verify or illustrate more about the reasoning behind adopting such an assumption?**”
>
> Our method relies on an augmented version of the label shift assumption. The fact that we get better performance on a variety of datasets is empirical demonstration of the reasonableness of our assumption. However, we do not claim that it captures all (or even most) forms of distribution shift - it depends on the richness of the latent factors z.
>
> “**In Figure 2, can the authors explain more about the U-shape performance of TTLSA? Specifically, why the performance of TTLSA under shift parameter (0.0) might be worse than that of shift parameter (1.0)?**”
>
> Performance for 0.0 is in fact the same as for 1.0, modulo experimental noise. To see why, note that if there is 100% correlation  or anti-correlation between z and y, then it becomes easy to predict the class label y by simply learning to predict z and then setting y=z or y=not(z). Conversely, if there is no correlation (0.5 on x-axis), there is no shortcut to exploit, and the model reduces to the performance of ERM predicting p(y|x). This is why there is a symmetric U-shaped curve.
>
> “**It would be better if the authors could discuss the differences with two closely related works, i.e., [1, 2].**”
>
> We will add discussion of these papers.

---

> > ### Comment · Reviewer_wAGR · 2023-08-21
> >
> > Thanks authors for the responses. After reading the author's rebuttal and the other reviewers' comments, I would like to raise my score to (6: Weak Accept).

---

### Official Review · Reviewer_KyoC · 2023-07-07

**Soundness:** 3 good
**Presentation:** 4 excellent
**Contribution:** 3 good
**Rating:** 6
**Confidence:** 3

**Summary:**

- This paper tackles the test-time adaptation problem to mitigate spurious correlation. By assuming the pervasiveness of $p(x|y, z)$, the authors derived an alternative method for predicting the target distribution and proposed Test-Time Label-Shift Adaptation (TTLSA) to optimize it. The TTLSA involves two steps: training on the source with calibration and logit adjustment and adaptation step using the EM algorithm. The method is evaluated on Colored MNIST, CheXpert, and four popular worst-group benchmark datasets, demonstrating superior adaptation performance (Figure 2) and compatible benchmark performances (Table 2).

Minor corrections: Some references to figures are presented as double question marks (??).

**Strengths:**

- The paper has a coherent flow, making it easy to follow, and the details and evidence are well-presented.
- The motivation for test-time adaptation is solid, and the experimental results in Figure 2 provide strong support.
- The ERM performances in Table 2 are reliable, as they closely match the accuracies achieved in [1].

[1] Idrissi, Badr Youbi, et al. "Simple data balancing achieves competitive worst-group-accuracy." *Conference on Causal Learning and Reasoning*. PMLR, 2022.

**Weaknesses:**

- One of my concerns is the assumption of having an annotated training dataset. The field is evolving towards building algorithms without relying on group information.
- Another concern is in comparison with Group DRO. Both methods assume that group information is available in the source dataset. TTLSA shows better test accuracy in this experiment, but not for worst-group accuracy.
- The Group DRO performance is lower than what is presented in [2].

[2] Izmailov, Pavel, et al. "On feature learning in the presence of spurious correlations." *Advances in Neural Information Processing Systems* 35 (2022): 38516-38532.

**Questions:**

In [2], it is claimed that Group DRO performs well because it improves the last linear layer rather than the underlying feature representations. How does TTLSA address this aspect?

**Limitations:**

The proposed method requires group information in the training dataset, which can be costly.

---

> ### Author Rebuttal · Authors · 2023-08-09
>
> Thank you for your review. Response to your points is below.
>
> “**One of my concerns is the assumption of having an annotated training dataset.**"
>
> We agree this is a weakness. However, as we show in Appendix C.5 (in supplemental), it is possible to apply our method with a very small amount of labeled data. In particular, we get good results on the worst group benchmarks even when up to 90% of the z labels are missing. The trick is to use label imputation to infer the missing z labels before applying our algorithm. We will clarify this in the revised paper.
>
> “**TTLSA shows better test accuracy in this experiment [than gDRO], but not for worst-group accuracy.**”
>
> In table 2, our worst group performance is actually on par with gDRO for 3 out of 4 datasets, with the exception being CelebA. For example, on Waterbirds, gDRO gets 87.98 and we get 88.38. However, the main focus of our paper is on average performance.
>
> “**The Group DRO performance is lower than what is presented in Izmailov**”.
>
> We ran the gDRO code by Idrissi (with their published hyper-parameters) and reported our findings, which agrees with the original version by Sagawa. Recently, Izmailov provides another implementation of gDRO, but we did not use their code, nor do we know why they reported different numbers for their baseline.
>
> “**In [2], it is claimed that Group DRO performs well because it improves the last linear layer rather than the underlying feature representations. How does TTLSA address this aspect?**”
>
> The purpose of TTLSA is to make better predictions in the presence of target label shift, not learning better representations. In fact, both logit adjustment and TTLSA operate ad hocly on the output logits, and are thus oblivious to the model internals (which is why it applies to both neural networks and gradient boosted trees). That being said, asking the model to predict (y, z) jointly instead of only y may force it to learn a better presentation for features relevant to z.

---

> > ### Comment · Reviewer_KyoC · 2023-08-20
> > **Response to authors**
> >
> > Thank you for the detailed response. I will retain my original score.

---

### Author Rebuttal · Authors · 2023-08-09

We respond to each reviewer separately below. However, a common concern was our dependence on a training set which has y and z labels. We agree this is a weakness. However, as we show in Appendix C.5 (in the supplemental material), it is possible to apply our method with a very small amount of labeled data. In particular, we get good results on the worst group benchmarks even when up to 90% of the z labels are missing. The trick is to use label imputation to infer the missing z labels before applying our algorithm. We will clarify this in the revised paper. (Note that the theoretical work in Yong'22 shows that some amount of environment labeling is required, absent further assumptions.)

L. I. N. Yong, S. Zhu, L. Tan, and P. Cui, “ZIN: When and How to Learn Invariance Without Environment Partition?,” in Advances in Neural Information Processing Systems, May 2022 [Online]. Available: https://openreview.net/forum?id=pUPFRSxfACD.

---

### Decision · Program_Chairs · 2023-09-21

**Decision:**

Accept (poster)

**Comment:**

The rebuttal has (at least partially) addressed the reviewers' concerns and all reviewers have reached an agreement to accept the paper. Therefore, I would recommend acceptance of the paper. Please revise the paper accordingly in the camera-ready version.